# The Dark Triad Traits of Firefighters and Risk-Taking at Work. The Mediating Role of Altruism, Honesty, and Courage

**DOI:** 10.3390/ijerph18115983

**Published:** 2021-06-02

**Authors:** Andra Cătălina Roșca, Vlad Burtăverde, Cristina-Ioana Dan, Alexandru Mateizer, Carol Robert Petrancu, Adrian Ionuț Iriza, Cristina Adina Ene

**Affiliations:** 1Faculty of Political Sciences, National University of Political Studies and Public Administration, 012104 Bucharest, Romania; catalina.rosca@politice.ro (A.C.R.); carol.petrancu@gmail.com (C.R.P.); 2Faculty of Psychology and Educational Sciences, University of Bucharest, 050663 Bucharest, Romania; alexandru.mateizer@gmail.com (A.M.); irizaadrianionut@yahoo.com (A.I.I.); cristina.adina.ene@gmail.com (C.A.E.); 3Centre for Psychosociology, Ministry of Internal Affairs, 023975 Bucharest, Romania; cristinaioanadan@yahoo.com

**Keywords:** risk-taking behaviors, Dark Triad traits, firefighters, life-history theory

## Abstract

Firefighting is considered a dangerous profession that imposes unique safety hazards. In this research, we investigated the relationship between the Dark Triad traits of firefighters (*N* = 1434, *Mage* = 39.03, *SD* = 6.9) and their risk-taking at work, considering the mediation role of altruism, honesty, and courage. We showed that firefighters high on Machiavellianism and psychopathy reported high risk-taking. Altruism, honesty, and courage mediated the relationship between Machiavellianism and risk-taking in firefighters. Honesty and courage mediated the association between psychopathy and risk-taking. Theoretical and practical implications are discussed.

## 1. Introduction

Firefighters work for the community aiming to protect it, preventing and managing fires. Their profession is made by activities that imply searching for missing persons (e.g., lost, drowned), providing first aid measures, extrication, unblocking/releasing, and saving people, animals, and property from fires and other disasters, and operate equipment in order to achieve these objectives. Firefighting is considered a dangerous profession that imposes unique health hazards [1]. Knowing how and why firefighters engage in risky behaviors and which degree of risk-taking is optimal in their profession, may help select candidates and also develop targeted interventions for managing the performance of firefighters (e.g., talent management settings).

Risk-taking refers to attitudes, activities, and behaviors that individuals engage in and are generally described as uncertain or dangerous [2]. Risk-taking is a component that is assessed by the most important error and accident management tools [3]. In dangerous professions, as in the case of firefighters, the risk is embedded in the nature of their job. In this regard, to avoid accidents, errors, or hazards in dangerous jobs, there are safety protocols that the employees should follow when performing their activity [3]. As a consequence, firefighters should be courageous enough to be performant in their profession. Still, they must also conform to existing protocols and avoid unnecessary risks that may lead to hazards or negative outcomes in their activities. This is a paradoxical and contradictory aspect in the case of the firefighter’s profession. They must be attracted enough by risk to develop passion and commitment for their profession but not very much engaged in risk-taking in order to comply with the safety protocols of their job.

Considering that many of the individuals that become firefighters have vocational interests for this profession, as they are attracted to it, and vocational interests are associated with personality traits [4], we assume that personality traits may predict risk-taking in firefighters. Various personality dispositions are associated with risk-taking behaviors [5], the Dark Triad traits being among the strongest correlates of risk-taking [6]. As such, knowing the effects of the Dark Triad on risk-taking among firefighters may offer important knowledge in the area of the Dark side of personality at work.

The Dark Triad traits are psychopathy (i.e., impulsive, lack of empathy; [7]), narcissism (i.e., entitlement, attention-seeking; [8]), and Machiavellianism (i.e., desire to manipulate and deceive others; [9]). There are group and cultural differences in risk-taking in different risky situations [10]. The risk-taking attitude is defined as the willingness to exchange risk for a benefit that corresponds to it [11]. It seems that a person decides to take a risk due to differences in their perceptions of risks. As such, risk-taking is explained, to a great extent, by individual differences [7] and situational factors [2]. As we mentioned above, among the individual differences that explain risk-taking behaviors, we can find personality traits, such as the Dark Triad traits [7].

The link between the Dark Triad traits and risk-taking in firefighters can be understood by relying on Life History Theory [12]. Life History Theory is a mid-level evolutionary theory that describes the allocation of resources and energy to important life contexts by involving various tradeoffs [13]. Life History Theory assumes that the resources available to an organism at any given time are finite, and tradeoffs have to be made [14]. The allocation of resources is aimed at surviving, growth, and reproduction. The theory assumes that there are differences regarding the allocation of resources which exist on a continuum (r/K continuum) where one pole is represented by organisms that develop a fast strategy (r-selected traits), while the other pole is represented by organisms that develop a slow strategy (K-selected traits). The adopted strategies in the development of the organism aim to maximize fitness (through tradeoffs) by taking into account environmental conditions [14]. Organisms that develop in unstable (e.g., scarce resource availability) and unpredictable environments (e.g., high physical risks, predation, etc.) will develop r-selected traits that are clustered together and form a fast life history strategy [15]. Contrary, organisms that develop in stable and predictable environments tend to develop K-selected traits that cluster together and form slow life-history strategies. Childhood environments characterized by harshness and unpredictability lead to individuals developing a fast life history strategy [16] which is associated with early physical maturation and giving birth at a young age [15]. Developing a fast life history strategy, which implies early reproduction, offers an evolutionary advantage as harsh environments mean the life span is shorter [17]. The behavioral characteristics of individuals characterized by fast life-history strategies imply the orientation on short-term gains and an opportunistic lifestyle, sexual variety, little parental investment, disregard for social rules, little social support, and extensive risk-taking [15].

In contrast, environments characterized by less harshness determine individuals to adopt a slow life history strategy as in such environments, life span is longer and allows individuals to invest in their personal and social development and in their offspring [15]. The behavioral characteristics of individuals characterized by slow life history strategies imply long-term planning, monogamy, compliance with social rules, high parental investment, and risk avoidance [15]. Consequently, researchers showed that the Dark Triad traits are behavioral dispositions that reflect fast life-history strategies, as individuals high on the Dark Triad traits are characterized by an agentic lifestyle, impulsivity, and disregard for social rules [18].

Fast life history strategies predict a variety of socially deviant behavioral outcomes such as gambling, risky sexual activity, drinking, smoking, low self-control, and delinquent behaviors [15]. Drawing on the above presented theoretical framework, we predict that individuals who are high on the Dark Triad traits (fast history strategies) would report high risk-taking. This assumption is already supported by important empirical evidence. People high on the Dark Triad traits take risks due to their irregular lifestyles [19]. They are unable to adjust their impulses effectively and can take unnecessary risks for minimal gains [6,20,21,22].

To better understand and explain reality, behavioral researchers try to offer explanatory mechanisms for bivariate relationships as the one between the Dark Triad traits risk-taking behaviors in firefighting; as such, the researchers and practitioners can know if there are other variables that may affect this relationship and should be considered. In this way, the studied phenomenon can be understood as accurately as possible, and the ecological validity of the research should increase. Furthermore, to better understand the relationship between the Dark Triad traits and risk-taking, we must understand if this is a direct or indirect relationship. In this regard, we argue that courage, honesty, and altruism could mediate the relationship between the Dark Triad traits and risk-taking behaviors in firefighters.

We already mentioned that individuals with high levels of the Dark Triad traits engage in risk-taking. However, taking into account the complexity of human behavior, risk-taking should be explained by an interaction of multiple characteristics. One characteristic that may play an important role in the relationship between the Dark Triad traits and risk-taking in firefighters is courage. Researchers defined courage as a behavioral approach despite the experience of fear [23]. Individuals with high levels of psychopathy are oriented to thrill-seeking, impulsive, reckless, are not concerned with their physical integrity, and are low in anxiety [24]. Therefore, they are perceived by others as courageous [25]. Courageous individuals are fearless, brave, and attracted by situations that challenge them and elicit intense emotions [25]. As such, we expect that individuals high on courage frequently engage in risk-taking behaviors. Considering the above-mentioned rationale, we expect that psychopathy has an indirect effect on risk-taking behaviors through courage.

Another characteristic that may explain the relationship between the Dark Triad traits and risk-taking is honesty. Authors described honesty as avoiding manipulating others for personal gain, feeling little temptation to break the rules [26]. Further, individuals with high scores on the Dark Triad traits are low on honesty [27]. People high on honesty reported low risky behaviors, especially health-related risk and ethical risk activities [28]. Considering that we already predicted that the Dark Triad traits would predict risk-taking in firefighters, this relationship may be an indirect one through low honesty.

Finally, the last characteristic that we argue that may mediate the relationship between the Dark Triad traits and risk-taking in firefighters is altruism, which is defined as affection and concern for others [29]. First responders scored significantly higher than civilians on measures of psychopathy, fearlessness, boldness, heroism, and altruism [30]. As we mentioned above, individuals high on the Dark Triad traits engage in risk-taking [6,18,21]. Also, individuals with high scores on the Dark Triad traits are low on empathy, are high on entitlement, do not care about others, and are low on altruism [27,31]. Altruistic individuals are concerned about other people’s happiness and wellness [32]. Moreover, individuals high on altruism engage in prosocial behavior, are honest, and respect social rules. Therefore, individuals characterized by high altruism should not be opportunistic and impulsive. As such, we expect that firefighters high on altruism should be low on risk-taking behaviors. Consequently, the Dark Triad traits should indirectly affect risk-taking behaviors in firefighters through low altruism.

To sum up, in this research, we investigated the relationship between the Dark Triad traits and risk-taking behaviors in firefighters. We also tested the mediation effect of courage, honesty, and altruism on the Dark Triad traits and risk-taking relationship.

**Hypothesis 1** **(H1).**
*The Dark Triad traits of firefighters will be positively related to risk-taking at work.*


**Hypothesis 2** **(H2).**
*Altruism will mediate the relationship between the Dark Triad traits of firefighters and risk-taking at work.*


**Hypothesis 3** **(H3).**
*Honesty will mediate the relationship between the Dark Triad traits of firefighters and risk-taking at work.*


**Hypothesis 4** **(H4).**
*Courage will mediate the relationship between the Dark Triad traits of firefighters and risk-taking at work.*


## 2. Materials and Methods

### 2.1. Participants and Procedure

The participants were 1434 firefighters from 27 Romanian fire departments who were recruited by unit psychologists and invited to fill out a paper-and-pencil questionnaire on a voluntary basis. One of the authors is a psychologist within the Ministry of Internal Affairs (Fire Service being included). The invitation addressed to the participants contained the idea of voluntary participation in a psychological study. The participants were informed about the use of their anonymous responses and the required honesty of their contributions.

Unit psychologists administered all the measures after the informed consent of the participants was obtained. Unit psychologists are entitled to offer psychological assessment and support to the Fire Service (Fire unit) personnel. The provided psychological support covers different levels of intervention, from prophylactic measures (such as psychological preparedness and training) to fitness-for-duty evaluations and counseling/psychotherapy (when needed/asked for and if the psychologist has additional competencies in this area).

All the participants were informed about the data’s confidentiality and anonymity and the respondents were motivated to give sincere and open responses. Of all the participants, regarding their biological sex, 99.2% of respondents were men, and 0.8% (which roughly represents the percent of women firefighters in Romania) were women (1423 men and 11 women). The participants’ mean age was 39.03 years (SD = 6.9; range: 20–58 years). A total of 71.3% of the participants had a non-leadership position, and 28.7% had a leadership position. A rate of 61.5% held a high school diploma, and 38.5% had a university degree. The majority of participants were married or in a relationship (81.6%), while 17.7% were single or divorced. Regarding the level of seniority, 68.8% of the participants were in service for 10 to 15 years, 11.2% were in service for more than 20 years, 3.3% were in service for 5 to 10 years, and 7.9% from 0 to 5 years.

### 2.2. Measures

The Dark Triad traits. The Dark Triad traits were measured using the Dirty Dozen [33]. This measure consists of 12 items that measure Machiavellianism (a = 0.83), narcissism (a = 0.70), and psychopathy (a = 0.54) (four items for each trait), each of them were scored on a seven-point Likert scale (1 = totally disagree; 7 = totally agree). The index of each trait was obtained by averaging the four corresponding items.

Risk-taking. We measured risk-taking using the IPIP version of the Jackson Personality Inventory Risk-Taking Scale [34], which consists of 10 items that measure risk-taking (i.e., I take risks, I seek danger) and adapted it to the work context. In this regard, we added the following frame of reference—“at work” [35] at the end of each item (e.g., I take risks at work, I seek danger at work). Each item was scored on a five-point Likert scale (1 = totally disagree; 5 = totally agree). We obtained the index of risk-taking by averaging all the 10 items (a = 0.70).

Altruism. We measured altruism using the IPIP version [34] of the NEO PI-R subscale of altruism, consisting of 10 items (e.g., Love to help others, Am concerned about others), each of them scored on a five-point Likert scale (1 = totally disagree; 5 = totally agree). We obtained the index of altruism by averaging all the 10 items (a = 0.73).

Honesty. We assessed honesty using the IPIP scale [34], which consists of 12 items (e.g., Keep my promises, Believe that honesty is the basis for trust), scored on a five-point Likert scale (1 = totally disagree; 5 = totally agree). The index of honesty was obtained by averaging all the 12 items (a = 0.70).

Courage. We measured courage with the self-perceived courageousness scale [36], which consists of 12 items (e.g., Even if I feel terrified, I will stay in that situation until I have done what I need to do, If there is an important reason to face something that scares me, I will face it), scored on a five-point Likert scale (1 = totally disagree; 5 = totally agree). The index of honesty was obtained by averaging all the 12 items (a = 0.72).

### 2.3. Statistical Analysis

Before running the inferential statistic procedures, we performed a series of data inspection procedures. We inspected the univariate distribution of all variables relying on Skewness and Kurtosis indicators. The values suggested a normal univariate distribution for all variables (Skewness and Kurtosis values between −2 and +2). To test the associations between the Dark Triad traits and risk-taking at work we used Pearson bivariate correlations. To investigate the predictive power of the Dark Triad traits on risk-taking we used hierarchical linear regression. We used t independent sample test to investigate sex differences for the study variables. To test the mediating effect of altruism, honesty, and courage on the relationship between the Dark Triad traits and risk-taking at work we used the medmod package for R and Jamovi to perform moderation analysis.

## 3. Results

Table 1 presents the means, standard deviations, and bivariate correlations between all the research variables. We can see that firefighters high on Machiavellianism were low on altruism, courage, honesty, and high on risk-taking. Firefighters with high levels of narcissism were low on altruism and honesty and high on risk-taking. Firefighters with high scores on psychopathy were low on altruism, courage, and honesty and high on risk-taking. Men were higher on psychopathy and risk-taking than women. There were no differences in the Dark Triad traits, altruism, honesty, courage, and risk-taking between firefighters in terms of year of service categories.

We used hierarchical linear regression (Table 2) to test the Dark Triad traits’ predictive power on risk-taking at work in the case of firefighters (Table 2). We found that the Dark Triad traits significantly predicted risk-taking at work in the case of firefighters (*R*2 = 0.05; *F*[3, 1431] = 21.20, *p* = 0.002). The residuals of narcissism (*β* = 0.07, *p* = 0.012) and psychopathy (*β* = 0.16, *p* = 0.002) predicted risk-taking at work. Further, we ran a mediation analysis to test the mediation role of altruism, courage, and honesty on the relationship between the Dark Triad traits and risk-taking at work in the case of firefighters.

To test for mediation, we used the medmod package for R and Jamovi to perform moderation analysis (see Table 3 and Figure 1, Figure 2, Figure 3, Figure 4 and Figure 5). The mediation analyses were performed using the bootstrapping method with bias-corrected confidence estimates.

We can see that Machiavellianism was negatively related to altruism (*β* = −23, *p* < 0.001). Altruism was negatively related to risk-taking (*β* = −0.07, *p* = 0.029). Machiavellianism was positively related to risk-taking (*β* = 0.09, *p* < 0.001). Further, controlling for altruism, Machiavellianism showed a weaker effect on risk-taking (*β* = 0.07, *p* = 0.003), suggesting a significant indirect effect and partial mediation. Further, Machiavellianism was negatively related to honesty (*β* = −0.16, *p* < 0.001). Honesty was negatively related to risk-taking (*β* = −0.12, *p* = 0.013). Machiavellianism was positively related to risk-taking (*β* = 0.09, *p* = < 0.001). Controlling for honesty, Machiavellianism showed a weaker effect on risk-taking (*β* = 0.07, *p* = 0.003), suggesting a significant indirect effect and partial mediation.

Machiavellianism was negatively related to courage (*β* = −0.17, *p* = < 0.001). Courage was positively related to risk-taking (*β* = 0.07, *p* = 0.004). Machiavellianism was positively related to risk-taking (*β* = 0.11, *p* = < 0.001). Controlling for courage, Machiavellianism showed a weaker effect on risk-taking (*β* = 0.09, *p* = 0.003), suggesting a significant indirect effect and partial mediation.

Psychopathy was negatively related to honesty (*β* = −0.13, *p* = < 0.001). Honesty was negatively related to risk-taking (*β* = −0.10, *p* = < 0.027). Psychopathy was positively related to risk-taking (*β* = 0.12, *p* = < 0.001). Controlling for honesty, psychopathy showed a weaker effect on risk-taking (*β* = 0.10, *p* = 0.003), suggesting a significant indirect effect and partial mediation. Finally, psychopathy was negatively related to courage (*β* = −0.16, *p* = < 0.001). Courage was positively related to risk-taking (*β* = 0.09, *p* = < 0.001). Psychopathy was positively related to risk-taking (*β* = 0.13, *p* = < 0.001). Controlling for courage, psychopathy showed a weaker effect on risk-taking (*β* = 0.11, *p* = 0.003), suggesting a significant indirect effect and partial mediation.

## 4. Discussion

In this research, we investigated the Dark Triad traits’ predictive power on risk-taking at work in firefighters. We also considered courage, honesty, and altruism as potential mediators in these relationships. We found that firefighters high on Machiavellianism and psychopathy reported high levels of risk-taking. These findings are congruent with our predictions and other research findings that identified a positive association between the Dark Triad traits and risk-taking [6,9]. A possible explanation of these findings may be that individuals high on the Dark Triad traits are characterized by fast life-history strategies [18], meaning that they have an agentic lifestyle, are high on impulsivity, and disregard social rules. All these characteristics concretize on behaviors such as gambling, risky sexual activity, gambling, drinking, smoking, low self-control, and delinquent behaviors. Considering the stability and the cross-situational manifestation of personality traits [35], we can infer that firefighters that are high on the Dark Triad traits manifest their fast life-history strategies on risky activities in their professional life as well, not only on their personal and social life.

It is important to mention that firefighters that were high on narcissism did not report high risk-taking. This is partially sustained by other research, which showed that narcissists engage in aggressive and risky behaviors only when they feel ego-threatened [6]. Considering that work activities in the field of dangerous professions rely on standard protocols [3], it is less likely that firefighters high on narcissism encounter situations at their job that should threaten their ego so as to make them act in an aggressive or risky manner.

We found that altruism partially mediated the relationship between Machiavellianism and risk-taking in the case of firefighters. Machiavellianism directly affected risk-taking, but there was also a significant indirect effect through low altruism, which suggested partial mediation. A possible explanation of this finding may be that firefighters that are high on Machiavellianism engage in risky behaviors at their job due to their low concern for others and their desire to manipulate, which are reflected in low altruism. This idea is partially sustained by other research that found that individuals high on Machiavellianism tried to look altruistic to others when they wanted to be put in a favorable position [37]. Considering the highly standardized and routinized nature of firefighters’ jobs, those high on Machiavellianism do not have any intention to look altruistic to obtain favorable positions.

Further, honesty partially mediated the association between Machiavellianism and risk-taking in firefighters. Machiavellianism exerted a direct effect on risk-taking, but there was also a significant indirect effect through low honesty, which suggested partial mediation. We can explain this finding considering that firefighters high on Machiavellianism perform risky activities at work because they disregard prosocial behavior and social rules, characteristics that are specific to low honesty [26]. This assumption is sustained by the negative association between Machiavellianism and honesty [27].

Courage mediated the association between Machiavellianism and risk-taking at work in firefighters. Machiavellianism exerted a direct effect on risk-taking, but there was also a significant indirect effect through low courage, suggesting partial mediation. We can explain this finding by relying on the idea that firefighters high on Machiavellianism avoid risky activities at their job through low courage, which may be due to the fact that Machiavellians are cautious in their actions if those actions lead to personal losses [8] as it is the case of risky behaviors at work.

Honesty mediated the relationship between psychopathy and risk-taking at work in firefighters. Psychopathy directly affected risk-taking, but there was also a significant indirect effect through low honesty, suggesting partial mediation. A possible explanation of this result may be that firefighters high on psychopathy perform risk-taking activities at work because they are high on impulsivity and have an opportunistic lifestyle, which is characteristic of low honesty. This idea is supported by the negative relationship between psychopathy and honesty [8].

Courage mediated the association between psychopathy and risk-taking in firefighters. Psychopathy directly affected risk-taking, but there was also a significant indirect effect through low courage, which suggests partial mediation. Even if individuals high on psychopathy are perceived as courageous by others [25], in our research, contrary to our predictions, there was a negative link between psychopathy and courage in firefighters. We may infer for individuals high on psychopathy that even if they are high on thrill-seeking and impulsivity [7], when it comes to behaviors that they manifest at their job, they do not act in a courageous manner because they may perceive that work behaviors do not bring them personal rewards and may avoid risking their psychical integrity for activities that offer personal benefits to others or the community. This idea is supported by the fact that individuals high on psychopathy engage in thrill-seeking activities that give them intense emotions and immediate rewards [8]. As such, firefighters with high levels of psychopathy are less courageous at their job, which makes them avoid risky behaviors.

Authors [38] argue that the decisions firefighters are making in a field marked by uncertainty, complexity, and high risk are defined by the judgment between risk (in terms of personal cost) and benefit. Within this framework, this study’s results could be explained by the particular manner in which Machiavellianism and psychopathy traits impact individual appraisal of costs, benefits, and deficits. This might also be one of the reasons behind barrier crossings in avoiding, cheating, or violating standard procedures and requirements in the workplace.

The indicators that reflect that a society can be described as a stable versus harsh environment (and, therefore, its citizens should be high on slow or fast life-history strategies) are resource availability, poverty, physical safety [15]. However, modern societies have changed drastically [39], and indicators that indirectly affect surviving (that affect adaptation and the quality of life) should be included as well. Such indicators are educational and medical infrastructure, equality of chance and transparency in public decision-making, a balanced labor market, and corruption [40]. Romania is one of the countries most affected by corruption in the European Union [41]. It has one of the lowest minimum wages [42], and one of the highest rates of unemployment [43]. Relying on this evidence, we can assume that Romania can be considered a harsh environment from an evolutionary perspective, where fast life-history strategies may have been adapted over time, and, as a consequence, Romanian firefighters may be higher on fast life-history strategies compared to firefighters from Western countries. This state of affairs may affect our findings as well. Firefighters from Romania may be higher on fast life-history strategies, and implicitly on the Dark Triad traits, which may explain the positive association between the Dark Triad traits and risk-taking at work in firefighters. Further studies should try to replicate the relationship between the Dark Triad traits and risk-taking in firefighters in western societies where individuals are higher on slow life-history strategies.

Our research has some important practical implications. Relying on the findings of this research, psychologists and practitioners involved in the recruitment and selection process might take into account the predictors of risk-taking, such as Machiavellianism and psychopathy. By doing this they may enhance the safety behaviors of firefighters by considering those candidates who are low on risk-taking at work. During the pre-employment psychological screening, personality investigation could be enriched by adding Machiavellianism and psychopathy tools in order to screen out candidates prone to risk-taking behaviors. In order to do that, additional research is needed to assess the actual incremental validity of Machiavellianism and psychopathy over other personality factors in predicting safety behaviors at work.

More than that, organizational development specialists or unit psychologists may devise training programs that target the honesty, altruism, and courage of firefighters with the aim of modulating their risk propensity in accordance with job characteristics and demands. As a part of prophylactic psychological assistance, psychological training programs target theoretical preparedness (where firefighters are made aware of psychological aspects impacting their work) and/or practical preparedness (through small group training, exercises aiming at developing attitudes, skills, abilities required for better performance at the job, at individual and/or group level).

Despite the fact that our research is the first one that investigated the role of the Dark Triad traits of firefighters in predicting risk-taking, it has a series of limitations. Firstly, our research is cross-sectional, which cannot permit us to make causal inferences. Further studies may try to investigate the topic of the research relying on an experimental research design. Secondly, we measured risk-taking relying on a trait measure, which does not capture the influence of various work situations that may influence risk-taking in firefighters. Future studies should measure risk-taking relying on a job simulation or a scenario setting, which allows identifying the role that critical situations play in risk-taking behaviors at work. Thirdly, we measured all the variables relying on self-report measures. Researchers showed that self-report methods could capture individual difference variance associated with the construct under consideration [44]. For example, in our research, courage, honesty, and risk-taking may have important demand characteristics associated with them in the case of firefighters. Even if all the answers of the participants were anonymous, considering the nature of their job, it would be difficult to admit that they are not courageous or they are high in risk-seeking. Further research should use objective behavioral outcomes to measure these characteristics. Fourthly, further research should measure the Dark Triad traits with a more comprehensive measure to avoid construct underrepresentation and low internal consistencies.

To conclude in this research, we investigated the predictive power of the Dark Triad traits on risk-taking at work in firefighters and tested the mediation effect of altruism, honesty, and risk-taking. We showed that the Dark Triad traits predicted risk-taking at work in firefighters, while altruism, honesty, and courage mediated the relationship between Machiavellianism and risk-taking. Also, honesty and courage mediated the relationship between psychopathy and risk-taking at work. Despite the above-mentioned limitations, this is the first research that investigated the role of the Dark Triad traits in predicting risk-taking at work in firefighters, which should represent a starting point for other research on the general topic of the Dark side of personality at work.

## Figures and Tables

**Figure 1 ijerph-18-05983-f001:**
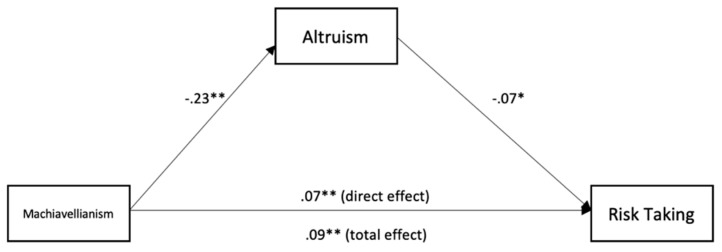
Mediation analysis for the relationship between Machiavellianism, altruism, and risk-taking. * *p* < 0.05; ** *p* < 0.01.

**Figure 2 ijerph-18-05983-f002:**
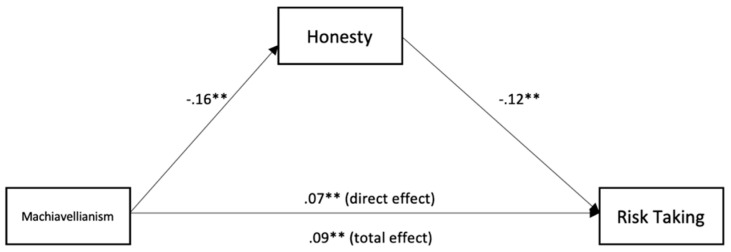
Mediation analysis for the relationship between Machiavellianism, honesty, and risk-taking. ** *p* < 0.01.

**Figure 3 ijerph-18-05983-f003:**
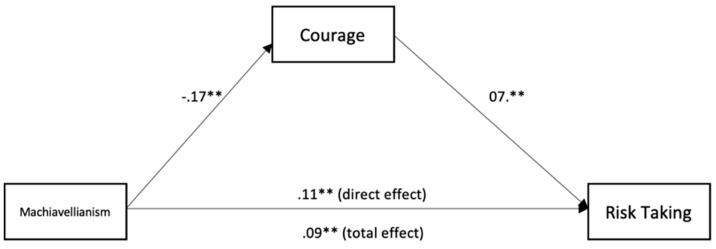
Mediation analysis for the relationship between Machiavellianism, courage, and risk-taking. ** *p* < 0.01.

**Figure 4 ijerph-18-05983-f004:**
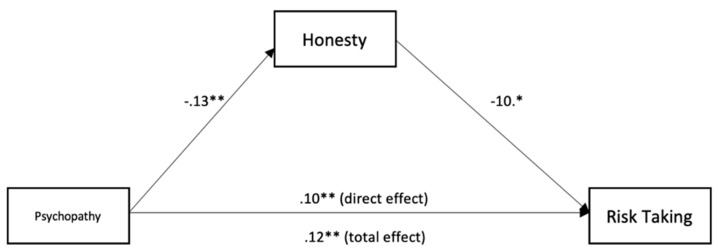
Mediation analysis for the relationship between psychopathy, honesty, and risk-taking. * *p* < 0.05; ** *p* < 0.01.

**Figure 5 ijerph-18-05983-f005:**
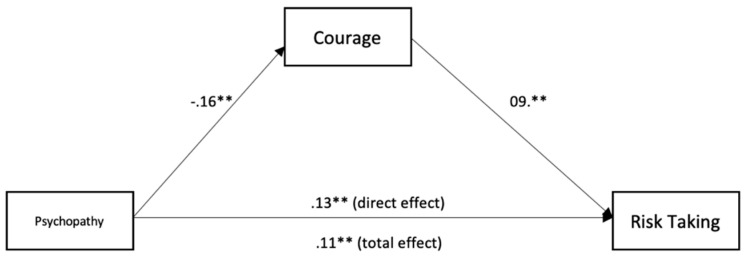
Mediation analysis for the relationship between psychopathy, courage, and risk-taking. ** *p* < 0.01.

**Table 1 ijerph-18-05983-t001:** Means, standard deviations, and bivariate correlations between all study variables.

Variable	1	2	3	4	5	6	7	*M*	*SD*
1. Machiavellianism	-							1.66	0.83
2. Narcissism	0.41 **	-						2.99	1.18
3. Psychopathy	0.61 **	0.30 **	-					2.10	0.87
4. Altruism	−0.42 **	−0.08 **	−0.36 **	-				4.54	0.45
5. Courage	−0.27 **	−0.01	−0.25 **	0.49 **	-			4.08	0.54
6. Honesty	−0.44 **	−0.08 **	−0.36 **	0.70 **	0.48 **	-		3.51	0.31
7. Risk-taking	0.15 **	0.13 **	0.19 **	−0.11 **	0.03	−0.12 **	-	2.83	0.53

** *p* < 0.01. Note: Machiavellianism, narcissism, and psychopathy were measured with the Dirty dozen, altruism was measured with IPIP version [34] of the NEO PI-R subscale of altruism, courage was measured with the self-perceived courageousness scale, honesty was measured with the IPIP scale of honesty, and risk-taking was measured using the IPIP version of the Jackson Personality Inventory Risk-Taking Scale.

**Table 2 ijerph-18-05983-t002:** Hierarchical linear regression on the predictive power of the Dark Triad traits on risk-taking at work.

Independent Variable	Risk-Taking			*LBCI*	*UPCI*
	*β*	*R* ^2^	*Δ R* ^2^		
Machiavellianism	0.02	0.12 **	0.12 **	−0.008	0.014
Narcissism	0.07 *			0.002	0.014
Psychopathy	0.16 **			0.014	0.034

* *p* < *0*.05; ** *p* < 0.01.

**Table 3 ijerph-18-05983-t003:** Mediation analysis regarding the role of altruism, honesty, and courage in the relationship between the Dark Triad traits and risk-taking at work in the case of firefighters.

Relationship	*B*	*SE*	*Z*	*p*
Machiavellianism ->Altruism	−0.23	0.01	−17.64	<0.001
Altruism ->Risk-taking	−0.07	0.03	−2.18	0.029
Machiavellianism ->Risk-taking (total effect)	0.09	0.16	5.59	<0.001
Machiavellianism ->Risk-taking (direct effect)	0.07	0.01	4.15	<0.001
Indirect effect	0.02	0.01	2.16	0.031
Machiavellianism ->Honesty	−0.16	0.01	−18.81	<0.001
Honesty ->Risk-taking	−0.12	0.04	−2.49	0.013
Machiavellianism->Risk-taking (total effect)	0.09	0.02	5.59	<0.001
Machiavellianism ->Risk-taking (direct effect)	0.07	0.01	3.91	<0.001
Indirect effect	0.02	0.01	2.46	0.014
Machiavellianism ->Courage	−0.17	0.02	−10.45	<0.001
Courage ->Risk-taking	0.07	0.03	2.88	0.004
Machiavellianism->Risk-taking (total effect)	0.09	0.02	5.59	<0.001
Machiavellianism ->Risk-taking (direct effect)	0.11	0.01	6.17	<0.001
Indirect effect	−0.02	0.001	−2.78	0.005
Psychopathy->Honesty	−0.13	0.01	−14.74	<0.001
Honesty ->Risk-taking	−0.10	0.04	−2.20	0.027
Psychopathy->Risk-taking (total effect)	0.12	0.01	7.43	<0.001
Psychopathy->Risk-taking (direct effect)	0.10	0.02	6.13	<0.001
Indirect effect	0.02	0.01	2.18	0.029
Psychopathy->Courage	−0.16	0.02	−9.87	<0.001
Courage->Risk-taking	0.09	0.03	3.29	<0.001
Psychopathy ->Risk-taking (total effect)	0.11	0.02	7.43	<0.001
Psychopathy ->Risk-taking (direct effect)	0.13	0.02	8.04	<0.001
Indirect effect	−0.013	0.004	−3.12	0.002

## Data Availability

Research data can be accessed at: https://drive.google.com/drive/u/0/search?q=faramiss (accessed on 1 March 2021).

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
