# Peer review of "The Dark Triad Traits of Firefighters and Risk-Taking at Work. The Mediating Role of Altruism, Honesty, and Courage"

_ijerph, 2021, doi:10.3390/ijerph18115983_

Round 1

Reviewer 1 Report

I thought your ideas were intriguing and I liked the theory supporting your ideas (Life History Theory).  However, I find the total use of self-report questionnaires as a design to be severely lacking in terms of value added to the scientific literature on this.  I am not universally opposed to self report measurement of construct...however, some constructs are challenging to get at with accuracy due to the nature of them.  Honesty is one.  Courage and risk-seeking are two that would have strong demand characteristics associated, particularly within a sample of firefighters.  Even though they knew that it was anonymous...firefighters are supposed to be courageous and not take unnecessary risks...it would be difficult to admit to not being very courageous as well as high in risk seeking.  

Also, the Dark Triad would be hard to get accurately from the participants themselves as the low mean value, particularly on Mach suggests.

Here are further detailed comments:

  • impact of culture, Romania, on findings?
  • what are "unit" psychologists
  • how did you gain access?  What was initial invitation - sample - and what was response rate? in addition to percentages for males and females, give actual numbers in parens 
  • psychopathy alpha is too low (Nunnally cut point standard is .60).
  • on Table 1, list scales in notes below so reader has context
  • I am not familiar with medmod package and Jamovi, I am more familiar with Hayes Process Macro for mediation and moderation...does your technique use bootstrapped samples for comparison...?
  • In limitations, absolutely discuss challenges of self-report and the demand characteristics as well as specifically the challenge of these particular constructs being measured with self report

Author Response

RESPONSE TO REVIEWER 1

Thank you for your suggestions and comments. We tried to keep in mind your recommendations when revising the article. We do consider that your observation increased the quality of the article.

  1. I thought your ideas were intriguing and I liked the theory supporting your ideas (Life History Theory).  However, I find the total use of self-report questionnaires as a design to be severely lacking in terms of value added to the scientific literature on this.  I am not universally opposed to self report measurement of construct...however, some constructs are challenging to get at with accuracy due to the nature of them.  Honesty is one.  Courage and risk-seeking are two that would have strong demand characteristics associated, particularly within a sample of firefighters.  Even though they knew that it was anonymous...firefighters are supposed to be courageous and not take unnecessary risks...it would be difficult to admit to not being very courageous as well as high in risk seeking.  

Also, the Dark Triad would be hard to get accurately from the participants themselves as the low mean value, particularly on Mach suggests.

Thank you for your positive feedback on our paper. Your comments were appreciated and we consider that your observations helped us enhance the quality of the paper. Indeed, relying only on self-report measures is an important limitation, especially when dealing with a population that is subjected to impression management and social desirability, as in the case of firefighters. However, considering the available resources, we were unable to rely on different measures. We added this as a limitation.

Here are further detailed comments:

  1. impact of culture, Romania, on findings?

Thank you for your suggestion. We added information about this in the Discussion section. p.8: The indicators that reflect that a society can be described as a stable versus harsh environment (and, therefore, its citizens should be high on slow or fast life history strategies) are resource availability, poverty, physical safety (Figueredo et al., 2006). However, modern societies have changed drastically (Neuberg & DeScioli, 2016), and indicators that indirectly affect surviving (that affect adaptation and the quality of life) should be included as well. Such indicators are educational and medical infrastructure, equality of chance and transparency in public decision making, a balanced labor market, and corruption (European Commission, 2020). Romania is one of the most affected countries by corruption from the European Union (European Commission, 2019). It has one of the lowest minimum wage (European Commission, 2021a), and one of the highest rates of unemployment (European Commission 2021b). Relying on this evidence, we can assume that Romania can be considered a harsh environment from an evolutionary perspective, where fast life history strategies may have been adaptive over time, and, as a consequence, Romanian firefighters may be higher on fast life history strategies compared to firefighters from Western countries. This state of affairs may affect our findings as well. Firefighters from Romania may be higher on fast life history strategies, and, implicitly on the Dark Triad traits, which may explain the positive association between the Dark Triad traits and risk-taking at work in firefighters. Further studies should try to replicate the relationship between the Dark Triad traits and risk taking in firefighters in western societies where individuals are higher on slow life history strategies. 

3. what are "unit" psychologists

Thank you for your observation. We added the description in the participants and procedure subsection: Unit psychologists are entitled to offer psychological assessment and support to the Fire Service (Fire unit) personnel. The provided psychological support covers different levels of intervention, from prophylactic measures (such as psychological preparedness and training), to fitness-for-duty evaluations and counselling/psychotherapy (when needed/asked for and if the psychologist has additional competencies in this area).

4. how did you gain access?  What was initial invitation - sample - and what was response rate?

Thank you for your observation. We added information about this in the participants and procedure subsection: One of the authors is a psychologist within the Ministry of Internal Affairs (Fire Service being included). The invitation addressed to the participants contained the idea of voluntary participation in a psychological study. The subjects were informed about the statistical use of their anonymous responses and the required honesty of their contributions. 

5. in addition to percentages for males and females, give actual numbers in parens 

 Thank you for your observation. We mentioned this in the participants and procedure subsection.

6. psychopathy alpha is too low (Nunnally cut point standard is .60).

Thank you for your observation. Indeed, the alpha for psychopathy is low. However, this is usually the case for measures composed of few items per scale, as in the case of the Dirty Dozen. We added this as a limitation.

7.on Table 1, list scales in notes below so reader has context

 Thank you for your suggestion. We added information about this.

8. I am not familiar with medmod package and Jamovi, I am more familiar with Hayes Process Macro for mediation and moderation...does your technique use bootstrapped samples for comparison...?

Thank you for your observation. Yes, the medmod package works exactly the same as Process. We added information about this in the results section.

9. In limitations, absolutely discuss challenges of self-report and the demand characteristics as well as specifically the challenge of these particular constructs being measured with self report

Thank you for your suggestion. We added this as a limitation.

Again, thank you for the suggestions and recommendations you made, as they were constructive information that helped us increase the paper's value.

Reviewer 2 Report

The authors present a manuscript examining the relationship between the Dark Triad traits of firefighters and their risk-taking at work, while considering the mediation role of altruism, honesty, and courage. 

This study is novel and provides great insight into the psychological characteristics of firefighters and how these may influence the firefighter's actions while on the job. 

MAJOR COMMENTS

The introduction is well-written and thorough. It provides substantial background on the various theories and variables to be examined. In the last paragraph, the authors summarize the objectives of the research study. This would be a good place to reiterate the study hypotheses as well. 

Materials and Methods

Line 153 – what is the percent of females in the Romanian fire services? IS the 0.8% in the study reflective of the composition in the fire service?

Line 153 – is this reported gender by the participant or reported biological sex?

Line 186 – there is no section on statistical analysis. There is minimal details of the statistical procedures used within the results section. A separate statistical analysis section is warranted to explain, in detail, which statistical approaches were used and how the data were analyzed.

Results

Can any of the main findings be visualized in graphical format? The results are impactful and presenting them in graph(s) may provide a larger impact to the reader. The correlation data in tables is useful but visualizing the variables that show relationships may help support the overall findings. 

Are there different responses between males and females?

If the data is grouped by years of services, are there any differences in the overall findings between groups? This may provide additional information on whether this is a fire service wide generalization or if one or more groups see the same responses and the other does not. 

Discussion

Lines 311-318 – can you provide additional information as to how firefighter candidates could be screened before getting hired? Could the assessment battery you used in this study be implemented as a pre-employment screening tool?

Lines 311-318 - Can you also expand on the “training programs” that could be implemented? Are these physical training programs or psychological training programs?

Line 326 – “should permit identify” is awkward. Please re-word.

Lines 327-329 – a conclusion paragraph would be more impactful than a concluding statement. The work the authors have produced is extensive and a full concluding paragraph is warranted to bring together the main findings and take home message.

MINOR COMMENTS

Lines 93 & 245 – gambling is listed twice

Line 229 – indent in the middle of sentence

Author Response

RESPONSE TO REVIEWER

Thank you for your suggestions and comments. We tried to keep in mind your recommendations when revising the article. We do consider that your observation increased the quality of the article.

The authors present a manuscript examining the relationship between the Dark Triad traits of firefighters and their risk-taking at work, while considering the mediation role of altruism, honesty, and courage. 

This study is novel and provides great insight into the psychological characteristics of firefighters and how these may influence the firefighter's actions while on the job. 

MAJOR COMMENTS

  1. The introduction is well-written and thorough. It provides substantial background on the various theories and variables to be examined. In the last paragraph, the authors summarize the objectives of the research study. This would be a good place to reiterate the study hypotheses as well. 

Thank you for your positive feedback on our paper. Your comments were appreciated, and we consider that your observations helped us enhance the quality of the paper. At your suggestion, we added the hypotheses at the end of the introduction.

Materials and Methods

  1. Line 153 – what is the percent of females in the Romanian fire services? IS the 0.8% in the study reflective of the composition in the fire service?

Thank you for your observation. Unfortunately, we do not have official information on this, but the unit psychologists that were involved in the process of Data collection said that the percent of female firefighters is about 1%.

  1. Line 153 – is this reported gender by the participant or reported biological sex?

Thank you for your observation. We clarified this by adding in the participants and procedure subsection the information we asked about their biological sex.

  1. Line 186 – there is no section on statistical analysis. There is minimal details of the statistical procedures used within the results section. A separate statistical analysis section is warranted to explain, in detail, which statistical approaches were used and how the data were analyzed.

Thank you for your observation. We added information about the statistical analyses that we used.

Results

  1. Can any of the main findings be visualized in graphical format? The results are impactful and presenting them in graph(s) may provide a larger impact to the reader. The correlation data in tables is useful but visualizing the variables that show relationships may help support the overall findings. 

Thank you for your suggestion. We added figures in the results section that present the findings for each significant relationship.

  1. Are there different responses between males and females?

Thank you for your observation. We tested for sex differences, and there were sex differences for psychopathy and risk-taking. However, considering that the groups were drastically unbalanced (1423 men and 11 women), the effect size is very weak.

  1. If the data is grouped by years of services, are there any differences in the overall findings between groups? This may provide additional information on whether this is a fire service wide generalization or if one or more groups see the same responses and the other does not. 

Thank you for your observation. There were no differences in the Dark Triad traits, altruism, honesty, courage, and risk taking between firefighters in terms of year of service categories.  We added this in the results section.

Discussion

  1. Lines 311-318 – can you provide additional information as to how firefighter candidates could be screened before getting hired? Could the assessment battery you used in this study be implemented as a pre-employment screening tool?

Thank you for your suggestion. We added information about this on the discussion section: During pre-employment psychological screening, personality investigation could be enriched by adding Machiavellianism and psychopathy tools, in order to screen-out the subjects prone to risk-taking behaviors. In order to do that, additional research is needed to assess actual incremental validity of Machiavellianism and psychopathy over other personality factors in predicting safety behaviors at work.

  1. Lines 311-318 - Can you also expand on the “training programs” that could be implemented? Are these physical training programs or psychological training programs?

Thank you for your suggestion. We added information about this in the discussion section: As a part of prophylactic psychological assistance, psychological training programs target theoretical preparedness (where firefighters are made aware of psychological aspects impacting their work) and/or practical preparedness (through small group training, exercises aiming at developing attitudes, skills, abilities required for a better performance at the job, at individual and/or group level). 

  1. Line 326 – “should permit identify” is awkward. Please re-word.

Thank you for your suggestion. We changed this with “should allow identify”.

  1. Lines 327-329 – a conclusion paragraph would be more impactful than a concluding statement. The work the authors have produced is extensive and a full concluding paragraph is warranted to bring together the main findings and take home message.

Thank you for your suggestion. We added a conclusion paragraph, as you suggested.

MINOR COMMENTS

  1. Lines 93 & 245 – gambling is listed twice

Thank you for your observation. We modified the text on this.

  1. Line 229 – indent in the middle of sentence

Thank you for your observation. We modified the text on this.

Again, thank you for the suggestions and recommendations you made, as they were constructive information that helped us increase the paper's value.
